# Opioid Pain Medication Prescription for Chronic Pain in Primary Care Centers: The Roles of Pain Acceptance, Pain Intensity, Depressive Symptoms, Pain Catastrophizing, Sex, and Age

**DOI:** 10.3390/ijerph17176428

**Published:** 2020-09-03

**Authors:** Carmen Ramírez-Maestre, Ángela Reyes-Pérez, Rosa Esteve, Alicia E. López-Martínez, Sonia Bernardes, Mark P. Jensen

**Affiliations:** 1Instituto de Investigación Biomédica de Málaga, Facultad de Psicología, Andalucía Tech, Universidad de Málaga, 29071 Málaga, Spain; angela_rp_@hotmail.com (Á.R.-P.); zarazaga@uma.es (R.E.); aelm@uma.es (A.E.L.-M.); 2Instituto Universitario de Lisboa (ISCTE-IUL), Cis-IUL, Av. das Forças Armadas, 1649-026 Lisboa, Portugal; sonia.bernardes@iscte-iul.pt; 3Department of Rehabilitation Medicine, University of Washington, 325 9th Ave, Seattle, WA 98104, USA; mjensen@uw.edu

**Keywords:** opioid prescriptions, pain acceptance, pain catastrophizing, depression, sex, chronic pain

## Abstract

*Background*: Psychological factors of patients may influence physicians’ decisions on prescribing opioid analgesics. However, few studies have sought to identify these factors. The present study had a double objective: (1) To identify the individual factors that differentiate patients who had been prescribed opioids for the management of chronic back pain from those who had not been prescribed opioids and (2) to determine which factors make significant and independent contributions to the prediction of opioid prescribing. *Methods*: A total of 675 patients from four primary care centers were included in the sample. Variables included sex, age, pain intensity, depressive symptoms, pain catastrophizing, and pain acceptance. *Results*: Although no differences were found between men and women, participants with chronic noncancer pain who were prescribed opioids were older, reported higher levels of pain intensity and depressive symptoms, and reported lower levels of pain-acceptance. An independent association was found between pain intensity and depressive symptoms and opioid prescribing. *Conclusions*: The findings suggest that patient factors influence physicians’ decisions on prescribing opioids. It may be useful for primary care physicians to be aware of the potential of these factors to bias their treatment decisions.

## 1. Introduction

Opioids have a selective affinity for central and peripheral opioid receptors and, via this mechanism, they can inhibit the transmission of nociceptive inputs and, therefore, the perception of pain [1]. Although opioids were originally formulated for acute and cancer pain, their use has expanded to treat a broad range of chronic pain conditions. However, recent research has highlighted the adverse effects of opioid medication due to the abuse of these drugs [2,3,4,5].

There is growing concern in Spain about the use and abuse of opioid analgesics [6,7]. In 2019, the Spanish Ministry of Healthcare published a report on medication use. The report analyzed the consumption of opioids based on prescription data provided by the Spanish National Health System. These data were obtained from private and public medical care centers, including hospitals. The results showed that the consumption of opioids in Spain had increased from 10.02 daily doses/1000 inhabitants/d in 2010 to 18.73 daily doses/1000 inhabitants/d in 2018.

This increase in opioid use has become a global phenomenon and is generating social concern. In a 2016 report, the International Narcotics Control Board (INCB) provided data on opioid abuse and its harmful effects. For example, a number of studies [8,9,10] have found that there is an increased mortality risk associated with long-term opioid therapy in individuals with chronic noncancer pain.

Although some physicians have previously suggested that opioids can benefit many patients [11,12], an increasing number of researchers have identified multiple adverse effects of long-term opioid use [2,3,13]. Their results are in line with the findings mentioned above. Moreover, new findings challenge the effectiveness of opioids in long-term chronic noncancer pain management [14,15]. Thus, despite growing evidence that opioids may be less than helpful for chronic pain management, the use of opioids for chronic pain management continues to increase [16].

Previous studies have shown that physicians’ judgments of chronic pain patients are influenced by variables pertaining to the patient, physician, clinical condition, and clinical encounter [17,18]. Several studies have sought to identify the factors that differentiate individuals with chronic noncancer pain who have been prescribed opioids from those who have not been prescribed opioids [19,20,21,22]. Sex and age are two demographic variables that have been identified as such factors. For example, in a literature review, Serdarevic et al. [19] suggested that the likelihood of being prescribed opioids was higher in women than in men. Similarly, the American Geriatrics Society guideline panel [20] and Frenk et al. [21] suggested that the likelihood of being prescribed long-term opioids was higher in elderly women than in men or younger women. Recently, Bedene et al. [22] found that one of the risk factors associated with opioid prescription was being more than 65 years old.

In addition, an association has been found between a number of pain characteristics and the likelihood of being prescribed opioids. Several international guidelines have emphasized the relevance of assessing pain intensity when making decisions on prescribing opioids [2,23,24,25,26,27]. These guidelines may underlie the finding that higher levels of pain intensity were reported by individuals who were prescribed long-term opioid therapy than individuals who were not prescribed such therapy [28]. However, pain experts have questioned the safety and clinical utility of focusing on pain intensity as the only or primary factor to consider when making decisions on prescribing opioids [29,30,31].

An association has been found between a number of psychological variables and chronic opioid prescription. For example, Adams et al. [32] noted that depressive symptoms are common among patients who have been initiated [33] and maintained on long-term opioid treatment [28,34,35]. Other studies have found that higher levels of depression have been reported among patients who were prescribed opioids compared to those who have not been prescribed opioids [22,36,37].

Pain catastrophizing is a variable that is closely associated with depression [38,39,40]. Pain catastrophizing has been defined as exaggerated maladaptive cognitions in response to ongoing, anticipated, or recalled pain [41]. Research has shown that this factor has a central role in chronic pain [42,43,44,45,46]. Finnan et al. [47] found that opioid use was higher among patients who catastrophized about their pain, even if they reported low levels of pain. Similarly, Jensen et al. [36] showed that catastrophizing was more frequent among opioid users than non-opioid users.

Pain acceptance is another psychological factor that has been studied as a predictor of opioid prescription. Pain acceptance involves not trying to control or avoid pain. One key component of pain acceptance is having the goal of engaging in valued activities regardless of pain [48]. Recently, Rhodes et al. [49] studied the associations between the acceptance and committed action components of the psychological flexibility model [50] and pain severity, pain interference, and the risk of opioid misuse in chronic pain patients. They found an association between higher levels of pain severity and pain interference and a higher risk of opioid misuse and that these associations were mediated by both acceptance and committed action. Similarly, Clementi [51] found an association between lower levels of pain acceptance and increased inpatient hospitalizations and more opioid and nonopioid prescription medications, while controlling for pain intensity, age, and sex [51]. An association has also been found between lower pain acceptance and psychological dependence on pain medications in individuals with chronic pain in the general population [52] and decreased opioid use in an addiction treatment setting [53].

Thus, a number of factors have been shown to differentiate patients who have been prescribed opioids from those who have not been prescribed opioids. These factors include sex, age, pain intensity, depression, pain catastrophizing, and pain acceptance. However, patients cannot prescribe opioids for themselves. Receiving an opioid prescription requires clinical interaction with a health care provider. Very few studies have sought to identify the factors associated with physicians’ decisions on prescribing opioid analgesics. Turk and Okifuji [37] studied the contribution of a set of factors that may influence the opioid prescribing practices of physicians for patients with chronic noncancer pain. The study was conducted in a tertiary care pain treatment center and included a sample of 191 individuals with heterogeneous pain disorders. The authors investigated the association between opioid prescribing by health care providers and a large set of patient variables, including sex, age, pain behavior, pain characteristics (i.e., pain severity, perceived life interference), the patient’s sense of control over their lives, and the patient’s levels of social support, pain disability, and depression. The results showed that patients prescribed opioids reported higher levels of depression and disability [37].

Sinnenberg et al. [54] conducted qualitative interviews with 52 emergency medicine physicians. These participants reported a range of factors in their decision-making, which included acuity, pain-related diagnosis, patient-reported pain severity, perceptions of the patient’s trustworthiness, and health system issues (i.e., concerns about patient satisfaction scores, hospital policy, regulatory environment, and guidelines)**.** In general, opioids were more likely to be prescribed to patients who were perceived to be honest than those perceived to be less honest. However, the authors did not ask the participants to describe how they determined this aspect [54].

Even less is known about opioid prescribing practices in the primary care setting [55]. In the only study we found on this topic, Tong et al. [56] found that long-term opioids were more likely to be prescribed to women than men and were more likely to be prescribed to patients who were more depressed than those who were less depressed. These results are in line with those on opioid use previously mentioned [19,20,21,37].

As Turk and Okifuji [37] recommended over 20 years ago, there remains a need for research on the factors associated with prescribing practices. Because most patients with chronic pain are treated by primary care physicians [57], knowledge regarding the factors associated with opioid prescribing in the primary care setting would be useful to identify the patients most likely to be prescribed opioids.

Given these considerations, the present study had a double objective: Firstly, to identify the individual factors that differentiate patients who had been prescribed opioids for the management of chronic back pain from those who had not been prescribed opioids and, secondly, to determine which factors make significant and independent contributions to the prediction of opioid prescribing when controlling for the other factors. No previous study has analyzed the effect of patients’ levels of pain acceptance on doctors’ decisions on prescribing opioids. Based on the findings from previous research, we hypothesized that the likelihood of being prescribed opioids would be higher in women and older patients than in men and younger patients. We also hypothesized that the likelihood of being prescribed opioids would be higher in patients who reported having higher levels of pain intensity, depressive symptoms, and pain catastrophizing and lower levels of pain acceptance than it would be in patients with lower levels of pain intensity, depressive symptoms, and pain catastrophizing and higher levels of pain acceptance. Given that these factors have not been analyzed within the same study as predictors of opioid prescription, we did not have any a priori hypotheses regarding which of these factors would make independent contributions to the prediction of opioid prescription.

## 2. Materials and Methods

### 2.1. Participants

The participants of this study had also been participating in an ongoing larger study designed to investigate the role of pain fear avoidance and pain acceptance in chronic pain adjustment (Project name: “Psychological variables involved in pain chronification” PSI2008-01803/PSIC; HUM-566, P07-SEJ-3067). The project was conducted in accordance with the Declaration of Helsinki and received ethical clearance by the Institutional Ethics Review Board (ERC UMA) and the Regional Hospital Ethics Committee. To date, two articles that have used data from this study have been published [44,58]. However, neither of these articles had the same objectives as the present study. The recruitment process lasted from May 2008 to January 2012. Individuals were considered eligible for inclusion if, at the moment of their participation in the study, they were experiencing back pain and had been experiencing pain for at least the last 3 months; they were not being treated for a malignancy, terminal illness, or psychiatric disorder; and they were able to understand the Spanish language. The physicians who participated in the study reviewed the patients’ clinical history. If the patients met the inclusion criteria, they were approached and invited to participate after completing the informed consent form. In total, 809 patients with back pain were invited to participate and 95 refused. The reasons for not participating were as follows: 15 patients did not reply to the phone calls; 63 stated they “had no time” for the assessment session; and 17 expressly refused participation. Of the patients initially contacted, 28 were excluded from the study because they did not meet the inclusion criteria or one of the exclusion criteria was present. Finally, 11 participants were also excluded because they gave us incomplete information. Thus, the final sample included 675 participants (see Figure 1). According to Austin and Steyerberg [59], a minimum subject-to-variable ratio of 5-to-1 in regression can produce stable and reasonably accurate estimates of the regression coefficients. Therefore, the sample size was excellent.

### 2.2. Variables and Instruments

#### 2.2.1. Demographic and Clinical Variables

Each participant was given a semi-structured interview with a psychologist to collect demographic, social, and medical history information.

#### 2.2.2. Pain Intensity

Patients were asked to rate their least, average, and worst pain during the past 2 weeks, as well as their current pain on a scale ranging from 0 to 10, with 0 indicating “No pain” and 10 indicating “Pain as intense as you could imagine.” A composite pain intensity score was calculated for each participant by calculating the average of these four ratings. Such composite scores have been shown to be very reliable measures of pain intensity in individuals with chronic pain [60]. In the present sample, internal consistency (Cronbach’s alpha) for the composite pain score was 0.84, indicating good reliability for this measure.

#### 2.2.3. Depressive Symptoms

The depression subscale of the Spanish version of the Hospital Anxiety and Depression Scale was used to assess depressive symptoms. This is a self-reporting scale that contains two 7-item scales, one for anxiety and one for depression [61]. The internal consistency of the depression subscale has been shown to be high (*α* = 0.86) in other samples [62]. In the present sample, internal consistency was 0.91, indicating excellent reliability.

#### 2.2.4. Pain Catastrophizing

A Spanish adaptation of the 13-item Pain Catastrophizing Scale (PCS) was used to assess pain catastrophizing [40,63]. Using this scale, respondents are asked to rate how often they experience catastrophizing thoughts and feelings when in pain [40]. The PCS can be used to assess three catastrophizing subdomains (rumination, magnification, and helplessness) that are strongly associated with one another. It also provides a total score. In the present study, all analyses used the total score. The Spanish Version of the complete PCS has been shown to have excellent internal consistency (Cronbach’s alpha = 0.95) [64]. In the present sample, internal consistency was 0.97, indicating excellent reliability.

#### 2.2.5. Pain Acceptance

We used the Spanish version of the 20-item Chronic Pain Acceptance Questionnaire (CPAQ-SV) [65,66] to assess pain acceptance. The CPAQ-SV yields a total score and two subscale scores, assessing Pain Willingness and Activity Engagement. The subscales of the CPAQ-SV have been shown to have adequate to good internal consistency (Activity Engagement, α = 0.85; Pain Willingness, α = 0.75) and good criterion validity [65]. In the present study, we used the total score as the measure of pain acceptance. In the present sample, internal consistency was *α* = 0.76, indicating adequate reliability.

### 2.3. Statistical Analysis

Descriptive statistics were computed for the sample and study variables. Continuous variables are expressed as the mean and standard deviations and categorical variables are expressed as numbers and rates. The chi-square test—for the categorical predictors—and t-tests—for the continuous predictors—were used to assess the hypothesized associations between the criterion variable of the study (i.e., whether or not the patient was prescribed opioids) and patient sex, age, patient-reported pain intensity, severity of depressive symptoms, pain catastrophizing, and acceptance. Cohen’s *δ* was also computed to assess the effect sizes of the associations. Values of Cohen’s *δ* between 0.20 and 0.50 indicate a small effect size, between 0.50 and 0.80 indicate a medium effect size, and more than 0.80 indicates a large effect size [67]. We examined the distributions of the study variables for normality using the Kolmogorov–Smirnov test to ensure that they met the assumptions for the planned regression analyses. To determine which predictors were independently associated with being prescribed opioids, we conducted a logistic regression analysis using opioid prescription (yes/no) as the criterion variable. In this analysis, all of the predictors were entered in a single step. Variables with *p* ≥ 0.05 were excluded from the equation. All data analyses were performed using the Statistical Package for the Social Sciences version 22.0 (SPSS; Chicago, IL, USA).

## 3. Results

### 3.1. Frequency Data for the Demographic and Clinical Variables

The participants had chronic back pain and had been attending four primary care centers. It was found that every participant had been given at least one prescription: 85 were prescribed opioids and 590 were prescribed nonopioid analgesics. Participants could indicate more than one site of back pain. Table 1 shows the participants’ characteristics.

### 3.2. Univariate Differences between Individuals Given and Not Given an Opioid Prescription

No statistically significant association was found between sex and opioid prescription (chi-square, *p* = 0.194).

Table 2 shows the results of the t-tests comparing differences between the two groups of participants. These results are shown by age, pain intensity, depressive symptoms, pain acceptance, and pain catastrophizing. As shown, significant effects were found for age (small effect), pain intensity (medium effect), depressive symptoms, and pain acceptance (small effects). However, no significant association was found between the two groups in pain catastrophizing.

### 3.3. Multivariate Associations between the Study Predictors and Differences in Opioid Prescribing

Pain catastrophizing was not included in the planned multivariate logistic regression analyses because no significant univariate association was found between this variable and differences in opioid prescribing (see Table 3). The logistic regression model was statistically significant (*χ*^2^ = 37.0101, *p* < 0.001) and explained 10% (Nagelkerke *R*^2^) of the variance in the criterion variable. The model correctly classified 87.3% of the cases. Pain intensity and depressive symptoms contributed significant unique variance to the prediction of the criterion variable.

## 4. Discussion

The first objective of this study was to identify the individual factors that differentiate patients who had been prescribed opioids from those who had not been prescribed opioids in primary care settings. We identified four patient variables (age, pain intensity, severity of depressive symptoms, and pain acceptance) that had a significant univariate association with opioid prescribing. Pain intensity and the severity of depressive symptoms were independently associated with being prescribed opioids. The findings are relevant to understanding why certain patients are prescribed opioids and to educating primary care providers on aspects that should be taken into consideration when making decisions on prescribing opioids rather than other treatments, such as antidepressants or multidisciplinary pain management.

Evidence suggests that men and women experience chronic pain in different ways [68]. Compared to men, women tend to report pain more frequently, refer to more bodily locations, describe longer pain duration [69], and report greater intensity [70]. Huhn et al. [66] found that self-reported pain ratings and scores on catastrophizing were higher in women with chronic pain who misused prescription opioids than those in men. Several studies have found similar differences between women and men in relation to catastrophizing [71,72,73]. Although deaths due to prescription opioid overdose are higher in men than in women [74], opioid-related overdoses have recently increased more in women than in men [75]. However, sex differences are not always found in this context. For example, Coloma et al. [76] analyzed prevalence in the use of and dependence on opioid drugs in a sample of Spanish individuals with chronic noncancer pain. The authors found no significant differences between men and women in the use (or abuse) of prescribed opioids. Overall, these findings suggest that there may be sex-based differences in prescribing opioids for chronic pain, although there is very little information on this aspect. Previous findings have suggested that the likelihood of prescription opioid use is higher in women than in men [18]. Furthermore, it has been found that the likelihood of being prescribed opioids was higher in women than in men [77]. However, our results are not consistent with these findings. The reasons for the inconsistent findings are not entirely clear. In general, inconsistent findings in the research suggest that contextual factors may have influenced the effect of a predictor on a criterion variable. This possibility has been supported by recent reviews, suggesting that gender-related biases in pain treatment are due to the characteristics of the patients as well as those of the health care providers [71,78]. Thus, opioid prescribing may depend on the sex of the patient—or their age (see below)—as well as on the sex of the health care provider (or their age, experience, training, etc). Overall, the inconsistent findings in the literature suggest the need for further research to identify the potential moderators of the role of the patient in relation to opioid prescribing.

Several studies have addressed age-related predictors of opioid abuse. Some have suggested that age does not play a role [79,80], whereas others have found that age is a risk factor in opioid prescription, with opioids being prescribed more often to older people than to younger patients [22,81]. Our findings suggest that, although patient age predicts opioid prescribing with a weak effect size, when it is analyzed as a univariate predictor, it is not significantly associated with opioid prescribing when other variables are controlled. This finding may help to explain the inconsistent findings reported in previous studies. Thus, when effects are weak, the statistical significance of a specific finding may depend on a variety of factors, such as the power (e.g., sample size) of the study, as well as the number of contextual factors related to the study population and setting. Overall, the findings suggest that age may play a role in physicians’ decisions to prescribe opioids and that they should therefore consider the potential influence of age-related biases in their decisions, particularly when factors other than patient age should be playing a more relevant role in these decisions. Given the inconsistent findings in the literature, which suggest the existence of moderating factors, research is needed to help elucidate the factors that could interact with patient age in order to determine which of these factors have a relevant role in opioid prescribing.

Our findings on patient-reported pain intensity as a predictor of opioid prescribing are consistent with those of previous researchers [28]. They are also consistent with older international guidelines that have recommended taking the patient’s reported pain intensity into account when making decisions about opioid prescribing [23,24,26]. However, these recommendations may now be outdated (see below). Pain intensity is a complex variable influenced by a larger number of biological, psychological, and social factors and is often assessed using rating scales [82]. The goal of analgesic treatment is to improve the patient’s ability to achieve functional goals [32], and thus physicians should not decide to prescribe opioids based solely on the results of pain intensity ratings [83]. Moreover, recent observational studies have found that long-term opioid therapy is associated with poor pain outcomes and greater functional impairment [84,85]. Contemporary guidelines state that opioids should be used only when the improvements in pain and function outweigh the expected risks [86]. Therefore, variables other than pain intensity alone should be taken into account when making decisions on prescribing opioids.

We also found an association between the severity of depressive symptoms and opioid prescriptions in our sample. Chronic pain and depression are closely related [87]. Depressive symptoms in chronic pain populations have an estimated prevalence between 30% and 50% [88,89,90,91]. These percentages are much higher than those in the general population [92,93]. Depression can worsen chronic pain, which may encourage opioid prescribing [87]. Moreover, experimental and prospective clinical studies have shown that opioids provide less pain relief in patients with chronic low back pain and depressive symptoms than they do in nondepressed individuals [94,95]. In line with the results of the present study, previous research has found that negative affect is a risk factor for opioid prescription misuse and abuse [96,97] and that individuals with chronic pain and comorbid depression are more likely to be prescribed opioids [22,98,99], prescribed higher doses of opioids [100], and prescribed opioids for longer periods [34]. These findings suggest that health care providers may inadvertently be prescribing opioids to treat depression, which in fact may make depression worse [90]. Sullivan suggested [87] that primary care physicians should be assessing depression in their patients before prescribing opioids. The Centers for Disease Control and Prevention Guideline for Prescribing Opioids for Chronic Pain (2016) recommended that physicians should ensure that treatment for depression is optimized and should consult with behavioral health specialists because treatment for depression can improve pain symptoms as well as depression. Therefore, the findings suggest that clinical practice may not be in line with current guidelines and that clinician knowledge concerning this issue should be updated as a matter of urgency. Indeed, if the objective of opioid treatment is to improve the patient’s ability to achieve functional goals, it would be relevant to conduct a psychological evaluation of the patients, rather than simply relying on reported pain intensity, to ensure that opioid treatment is effective in improving quality of life.

Previous studies have highlighted the relevance of pain acceptance in the adjustment of individuals to chronic pain [44,101]. Other studies have found an association between lower levels of pain acceptance and high medical utilization [51] and psychological dependence on pain medications [52]. However, we found no significant association between the participants’ reported levels of pain acceptance and opioid prescribing by primary care physicians. If this finding is replicated in future studies, it would suggest that health care providers may not need to screen patients’ levels of pain acceptance, which would typically be done to identify a factor that may bias their decision to prescribe opioids.

The detrimental role of catastrophizing in managing and coping with pain is well documented [42,43,44,45,46]. In a sample of patients with sickle cell disease, Finnan et al. [47] found an association between higher levels of pain catastrophizing and the increased use of short-acting opioids. Furthermore, Jensen et al. [36] found that catastrophizing was more often reported by opioid users than by nonopioid users. Given these findings, we expected to find differences in levels of pain catastrophizing between the participants who were prescribed opioids and those who were not. However, the findings did not support this hypothesis. This lack of significant associations may be due to there being something unusual about the study context or study sample. If so, catastrophizing may play a role in opioid prescribing in some settings but not in others. Alternatively, it may be the case that the level of catastrophizing plays a relevant role in opioid use (i.e., in patients who have a prescription), but does not necessarily play a role in the prescribing behavior of health care providers. Future research is needed to address these issues.

This study has a number of limitations that should be considered when interpreting the results. Firstly, the cross-sectional design of the study precludes the possibility of drawing causal conclusions regarding the associations found. Longitudinal methods would be useful to investigate the role of pain intensity and the severity of depressive symptoms in prospectively predicting the likelihood of chronic pain patients being prescribed opioids. Secondly, the study population was recruited from just four primary care centers and included individuals willing to participate in the larger ongoing survey study. Thus, they may or may not be representative of individuals with chronic pain in other settings. Additional research is needed to determine the extent to which the present findings can be replicated in other samples of individuals with chronic pain.

## 5. Conclusions

We found significant univariate associations between patient age, pain intensity, and depressive symptoms and opioid prescribing for individuals with chronic back pain attending primary care centers. In multivariate analyses, pain intensity and depression alone contributed significant unique variance to the prediction of opioid prescribing. However, physicians should not base their decision to prescribe opioids only on the patients’ reported levels of pain severity. Moreover, given that lower levels of pain relief with opioids are reported by patients with chronic pain and co-morbid depression than by patients without depression, physicians should consider whether treatments other than opioid medication may be more appropriate for patients with depression. These findings highlight the relevance of screening for depression in patients presenting with chronic back pain.

Primary care physicians write a large percentage of all opioid prescriptions, but little is known about the characteristics of patients who receive them. The findings from the present study emphasize the relevance of understanding all the variables related to opioid prescribing in primary care centers, rather than simply relying on reported pain intensity. Knowledge regarding these variables would be useful for identifying individuals who are more likely to be prescribed opioids. This knowledge could then be used to develop training programs for physicians to help them identify the patients most likely to be prescribed opioids, even when these analgesics may not be appropriate.

## Figures and Tables

**Figure 1 ijerph-17-06428-f001:**
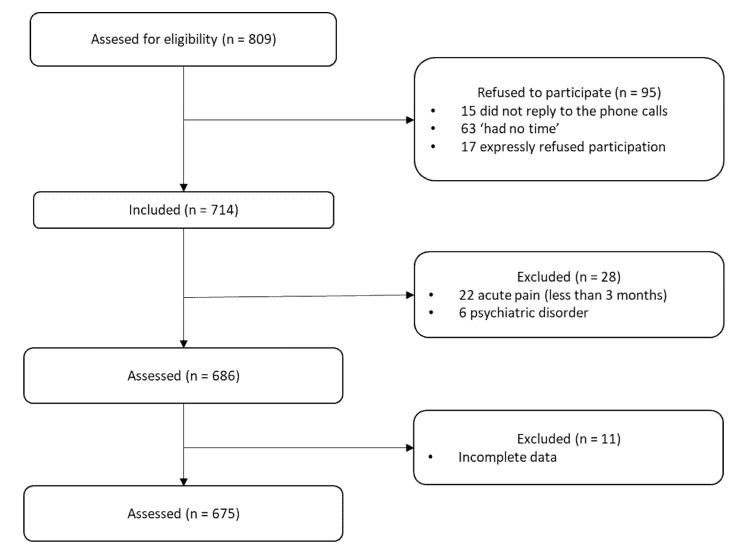
Flow chart of inclusion.

**Table 1 ijerph-17-06428-t001:** Frequency data for the demographic and clinical variables (*N* = 675).

Variables	Mean	SD	Min/Max
Age (years)	45.4	12.9	16/79
Time in pain (months)	49	59.3	4/240
Pain intensity	5.3	1.6	0.5/9.5
	*N*	*%*	
Sex			
Men	274	41	
Women	401	59	
Marital status			
Single	125	18.5	
Married	407	60.3	
Unmarried couple	54	8.0	
Divorced	39	5.8	
Separated	24	3.6	
Widowed	26	3.9	
Education			
Reading and writing	70	10.4	
Primary school	231	34.2	
High school	239	35.3	
University education	135	20.1	
Work status			
Housekeeping	112	16.6	
Working	352	52.2	
Studying	17	2.5	
Unemployed	102	15.1	
Retired	92	13.6	
Site of pain			
Upper back	435	64.4	
Sacrum	431	63.9	
Neck	361	53.5	
Lower back	251	37.2	

**Table 2 ijerph-17-06428-t002:** Differences between participants who were prescribed and not prescribed opioids.

	Participants Prescribed Opioids (*N* = 85)	Participants not Prescribed Opioids (*N* = 590)		*p* Value		Cohen’s *δ*
	Mean	SD	Mean	SD		
Age	49.04	11.05	44.88	13.12	0.002	0.34
Pain intensity	6.03	1.49	5.14	1.61	<0.001	0.57
Depressive symptoms	17.35	5.49	14.67	5.51	<0.001	0.49
Pain acceptance	65.49	7.97	69.60	9.96	<0.001	0.45
Pain catastrophizing	23.87	10.82	23.34	8.85	0.669	

**Table 3 ijerph-17-06428-t003:** Results of logistic regression analysis of variables associated with opioid prescribing.

	Beta	Wald	Exp(B)	*p*
Age	0.017	2.729	1.017	0.099
Pain intensity	0.245	9.206	1.278	0.002
Depressive symptoms	0.061	6.318	1.063	0.012
Pain acceptance	−0.013	1.027	0.987	0.311

Note: No prescription for opioids was coded as 0 and a prescription for opioids was coded as 1.

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
