# Peer review of "Opioid Pain Medication Prescription for Chronic Pain in Primary Care Centers: The Roles of Pain Acceptance, Pain Intensity, Depressive Symptoms, Pain Catastrophizing, Sex, and Age"

_ijerph, 2020, doi:10.3390/ijerph17176428_

Round 1
Reviewer 1 Report
Thanks to the authors for this interesting article.On line 99 remove the italics from not To improve the quality of the article, I would add a section on study design and sample size calculation
There are two sections with the name of the "participants". Change the name to one so as not to repeat information. Always put 0 in front of the decimal point in p values. In lines 264 and 270 it is necessary to put a period after "al" Section 5 name it as conclusions
I recommend checking all the bibliographic references well, since they do not adequately follow the Vancouver standards: When there is more than one author, remove the period after the initial of the name and leave only the comma. Do not put a period between the initials of an author's name. The initials must be continuous. As with the name of the journal, there is no point between each abbreviation of the journal title. There are references that do not have a period after the name of the journal or a semicolon after the year of publication.
Author Response
Comment 1
On line 99 remove the italics from
Answer
The italic form has been removed
Comment 2
To improve the quality of the article, I would add a section on study design and sample size calculation
Answer
Following the journal’s authors’ instructions, a study design section has not been included. However, we have included some information about sample size calculation (see in yellow, in participants section): According to Austin and Steyerberg [60] a minimum subject-to-variable ratio of 5-to-1 in regression can produce stable and reasonably accurate estimates of the regression coefficients. Therefore, sample size is excellent.
Comment 3
There are two sections with the name of the "participants". Change the name to one so as not to repeat information.
Answer
Thanks for noticing this mistake. We have deleted the “participants” at the beginning of Results’ section, although, following reviewer 3 suggestion, we have include a new table with the frequency data for the demographic and clinical variables.
Comment 4
Always put 0 in front of the decimal point in p values.
Answer
We have added the 0 number before p values (see in yellow).
Comment 5
In lines 264 and 270 it is necessary to put a period after "al"
Answer
A period has been included after “al.”
Comment 6
Section 5 name it as conclusions
Answer
Section 5 has been named “conclusions”
Comment 7
I recommend checking all the bibliographic references well, since they do not adequately follow the Vancouver standards: When there is more than one author, remove the period after the initial of the name and leave only the comma. Do not put a period between the initials of an author's name. The initials must be continuous. As with the name of the journal, there is no point between each abbreviation of the journal title. There are references that do not have a period after the name of the journal or a semicolon after the year of publication.
Answer
Thank you for your revision. This journal follows the ACS style (American Chemical Style). We have used Mendeley to include the references and, after that, we checked it following recent articles published in this journal (Vol 17, 2020). Again, thank you very much for your thorough review.
Reviewer 2 Report
First of all, I want to note that it has been a pleasure review your manuscript. I think this is an interesting topic for clinicians who manage this prevalent condition.
In order to improve the quality of the manuscript. After reading in depth the manuscript, I would like to make some comments and ask the authors several questions about.
Methods:
I would include a flow diagram
Línea 377: No incluiría esta referencia en las conclusiones porque es parte de la discusión
Author Response
REVIEWER 2
Comment 1
First of all, I want to note that it has been a pleasure review your manuscript. I think this is an interesting topic for clinicians who manage this prevalent condition.
Answer
Thank you very much for your comments
Comment 2
Methods: I would include a flow diagram
Answer
A Flow chart of inclusion has been included.
Comment 3
Línea 377: No incluiría esta referencia en las conclusiones porque es parte de la discusión
Answer
The reference has been removed. Thanks vry much for your review.
Reviewer 3 Report
Dear Authors,
I have some minor and major comments and suggestions about your paper about chronic pain.
Author Contributions should be presented before References – check Instructions for Authors;
Provide all details about Author List and Affiliations and correct the size of letters – check Instructions for Authors;
Check your Abstract due to recommendation in Instructions for Authors – abstract should be a total of about 200 words maximum and follow the style of structured abstracts with necessary information;
line 70 – you start the sentence with “Several studies…” and at the end of this sentence there is no references – you have to add;
line 90 – why in the bracket is information to see reference number 36 – why it is not with other references in bracket in line 89;
end of line: 124, 131, 136 should be reference;
Last paragraph in Introduction is very long but should present only the aim of your study, why or what was the reason do to this research – there is no need to present study sample, this information should be in section Materials and Methods;
Section Materials and Methods in part Participants give more details about the study group how many people took part and how many was rejected, add information about project identification code in context to ethical consideration, when and where this study was conducted, inclusion and exclusion criteria (you compare participants prescribed and not prescribed opioids think about criteria for this both groups) and all other necessary information about participant;
In Statistical Analysis there is no information about what kind of normality test was used and no information about p-value;
Why there is such big difference in groups size – your topic is about opioid pain medication prescription for chronic pain?
In section Results should be presented first table characterized participants of study;
In Conclusions you present that that chronic pain in patients of primary care centers, but in section Materials and Methods you inform that one of your criteria was only back pain for at least the past 3 months what about other problems of chronic pain?
Section 5 should be described as Conclusions due to Instructions for Authors;
Section References verify with Instructions for Authors;
Author Response
REVIEWER 3
Comment 1
Author Contributions should be presented before References – check Instructions for Authors;
Answer
We have removed “Author contributions” from the front matter.
Comment 2
Provide all details about Author List and Affiliations and correct the size of letters – check Instructions for Authors;
Answer
We have provided all details about Author List and Affiliations and corrected the size of letters. We are sorry about the mistakes.
Comment 3
Check your Abstract due to recommendation in Instructions for Authors – abstract should be a total of about 200 words maximum and follow the style of structured abstracts with necessary information;
Answer
Thank you. We have reduced the number of words of abstract and have included sections.
Comment 4
line 70 – you start the sentence with “Several studies…” and at the end of this sentence there is no references – you have to add;
Answer
Those studies are the ones explained below. The references have now been included.
Comment 5
line 90 – why in the bracket is information to see reference number 36 – why it is not with other references in bracket in line 89;
Answer
Thanks for noticing that. Now reference 36 is included in previous bracket.
Comment 6
end of line: 124, 131, 136 should be reference;
Answer
The references have been included
Comment 7
Last paragraph in Introduction is very long but should present only the aim of your study, why or what was the reason do to this research – there is no need to present study sample, this information should be in section Materials and Methods;
Answer
Following your suggestions, this paragraph has been reduced.
Comment 8
Section Materials and Methods in part Participants give more details about the study group how many people took part and how many was rejected, add information about project identification code in context to ethical consideration, when and where this study was conducted, inclusion and exclusion criteria (you compare participants prescribed and not prescribed opioids think about criteria for this both groups) and all other necessary information about participant;
Answer
A new paragraph (see in yellow) with more information about participants has been included.
Comment 9
In Statistical Analysis there is no information about what kind of normality test was used and no information about p-value;
Answer
We have added new information about the normality test used and about p-value in Statistical Analysis section (see in yellow)
Comment 10
Why there is such big difference in groups size – your topic is about opioid pain medication prescription for chronic pain?
Answer
We have a sample of 675 patients with chronic back pain that had been attending primary care centers. Of those, 14% were prescribed opioids, which is a high percentage. In fact, the percentages of opioid medication prescription in primary care centers in Spain use to be even lower nowadays (4% approx. according to Tormo Molina, J., Marín Conde, L., González Urbano, M., Ruiz Pérez de la Blanca, M., Robles Martín, J., & Vivar Simón, M. (2017). Prescripción de opioides mayores en pacientes con dolor no oncológico: descripción de sus características en una zona de salud de atención primaria. Revista de la Sociedad Española del Dolor, 24(1), 19-26). Therefore, the big difference in groups size is to be expected.
Comment 11
In section Results should be presented first table characterized participants of study;
Answer
The participants’ characteristics have been included in a new table (table 1), in results section.
Comment 12
In Conclusions you present that that chronic pain in patients of primary care centers, but in section Materials and Methods you inform that one of your criteria was only back pain for at least the past 3 months what about other problems of chronic pain?
Answer
We included this point in “limitations”. In line 358-359 we wrote: Additional research is needed to determine the extent to which the present findings can be replicated in other samples of individuals with chronic pain
Comment 13
Section 5 should be described as Conclusions due to Instructions for Authors;
Answer
We have change section 5’s name.
Comment 14
Section References verify with Instructions for Authors;
Answer
We have reviewed the references section. Thank you very much for your review.
Round 2
Reviewer 3 Report
Dear Authors,
Thank you for your replay.